# Nicotine enhances the stemness and tumorigenicity in intestinal stem cells via Hippo-YAP/TAZ and Notch signal pathway

**Ryosuke Isotani[1], Masaki Igarashi[1]\*, Masaomi Miura[1], Kyoko Naruse[1], Satoshi Kuranami[1], Manami Katoh[2,3], Seitaro Nomura[2,3], Toshimasa Yamauchi[1]\***

[1]Department of Diabetes and Metabolic Diseases, Graduate School of Medicine, The University of Tokyo, Tokyo, Japan; [2]Department of Cardiovascular Medicine, The University of Tokyo Graduate, School of Medicine, Tokyo, Japan; [3]Department of Frontier Cardiovascular Science, The University of Tokyo, Graduate School of Medicine, Tokyo, Japan

## eLife Assessment

This study presents a **valuable** finding on a potential signaling pathway responsible for the direct effects of nicotine on intestinal stem cell growth and tumorigenesis. The evidence supporting the claims of the authors is **solid**. This research will be of interest to medical biologists specializing in intestinal tumors.

**Abstract** Cigarette smoking is a well-known risk factor inducing the development and progression of various diseases. Nicotine (NIC) is the major constituent of cigarette smoke. However, knowledge of the mechanism underlying the NIC-regulated stem cell functions is limited. In this study, we demonstrate that NIC increases the abundance and proliferative activity of murine intestinal stem cells (ISCs) in vivo and ex vivo. Moreover, NIC induces Yes-associated protein (YAP) /Transcriptional coactivator with PDZ-binding motif (TAZ) and Notch signaling in ISCs via α7-nicotinic acetylcholine receptor (nAchR) and protein kinase C (PKC) activation; this effect was not detected in Paneth cells. The inhibition of Notch signaling by dibenzazepine (DBZ) nullified the effects of NIC on ISCs. NIC enhances in vivo tumor formation from ISCs after loss of the tumor suppressor gene Apc, DBZ inhibited NIC-induced tumor growth. Hence, this study identifies a NIC-triggered pathway regulating the stemness and tumorigenicity of ISCs and suggests the use of DBZ as a potential therapeutic strategy for treating intestinal tumors.

**\*For correspondence:**
migarashi@m.u-tokyo.ac.jp (MI);
tyamau@m.u-tokyo.ac.jp (TY)

**Competing interest:** The authors declare that no competing interests exist.

## Introduction

All tissues and organs are generated from stem cells. Abnormal stem cell functions are significantly associated with age-related organ dysfunction and carcinogenesis (*Adams et al., 2015*). Intestinal epithelial turnover is sustained by intestinal stem cells (ISCs) and adjacent Paneth cells. Paneth cells constitute the niche for ISCs that reside at the bottom of the crypts. Most of the ISCs are leucine-rich repeat-containing G-protein-coupled receptor 5 (Lgr5) +ones that regulate intestinal homeostasis in response to dietary signals (*Igarashi and Guarente, 2016*; *Yilmaz et al., 2012*). In the intestine, Lgr5 +ISCs are possibly the origin of precancerous adenomas (*Barker et al., 2009*). For instance,

**eLife digest** Cigarette smoking is one of the most significant and preventable health risks linked to a variety of diseases, including cancers. Cigarettes release over 5,000 chemicals when they burn, and at least 70 of them cause cancer.

Nicotine is a highly addictive component of cigarettes. Whilst it is not carcinogenic, it has been shown to affect various properties of many cell types, including stem cells. Stem cells function as a repair system and can be found in many tissues or organs. They can both self-renew and differentiate into specialized cell types. For example, intestinal stem cells (ISCs) are crucial for maintaining the structure of the intestines. However, when dysregulated they can lead to the development of tumors.

To find out how nicotine affects ISCs in mice, Isotani et al. exposed the animals and ISCs from mice grown in a laboratory to nicotine. Mice were fed 200 µg/ml (which emulates active smoking) for more than 8 weeks in their drinking water. They then used a technique known as immunohistochemistry as well as an organoid forming assay to study the behavior of the cells in the intestine of mice.

The results showed that exposure to nicotine caused ISCs in the small intestines of the treated mice to divide faster, which increased the number of these cells both in mice and in isolated cells. More specifically, nicotine activated a signaling pathway linked to cell growth, which caused the cells to multiply uncontrollably. When this signaling pathway was experimentally blocked, the ISCs were no longer overactive, and the mice did not develop any tumors.

In addition, Isotani et al. found that secretory cells, called Paneth cells, which are located near ISCs and have previously been shown to affect the growth rate of ISCs, did not show any changes to these pathways and their cell growth. This suggests that changes to the pathways affecting ISCs are not linked to Paneth cells.

To develop therapeutic strategies against nicotine-related diseases, including intestinal tumors, it is essential to understand how nicotine influences ISCs behavior. The results highlight the critical role of specific pathways in regulating the growth of ISCs and the formation of intestinal tumors linked to smoking. A next step would be to validate the therapeutic potential of drugs that can inhibit pathways linked to ISC growth.

---

long-term high-fat diet (HFD)-induced obesity enhances the self-renewal potential of ISC and promotes in vivo formation of tumors via the induction of peroxisome proliferator-activated receptor delta in ISCs (*Beyaz et al., 2016*).

Moreover, cigarette smoking has emerged as a potential major risk factor associated with colon cancer, along with metabolic risk factors such as diet and obesity. Previous studies indicate that cigarette smoking is significantly associated with colon cancer and mortality in humans (*Botteri et al., 2008*) and animal models (*Kim et al., 2008*).

Cigarette smoke contains a wide range of compounds that are harmful to human health; nicotine (NIC) derivatives 4-(methylnitrosamino)–1-(3-pyridyl)–1-butanone (NNK) and N'-nitrosonornicotine (NNN) are highly carcinogenic (*Brunnemann and Hoffmann, 1991*; *Schuller et al., 1995*), which can induce mutations in tumor suppressive genes like *Rasa*, *Trp53*, and *Rb* (*Sekido et al., 2003*). Although NIC itself, the addictive component in cigarette smoke, is generally considered to have a limited ability to initiate cancer, it can stimulate several effects crucial for cancer development independently (*Schaal and Chellappan, 2014*). However, knowledge of the mechanism underlying the NIC-regulated ISC functions and intestinal tumorigenicity is limited.

In this study, we aimed to demonstrate the effects of NIC treatment on the functions of murine ISCs ex vivo and in vivo. Moreover, we explored the molecular factors and signaling cascades prospectively associated with the regulation of the effects of NIC on ISCs. This study can potentially contribute to the comprehensive knowledge on the pathway by which stem cells respond to NIC, a major component of cigarette smoke, and suggest an effective new strategy for the treatment of smoking-related colon cancer.

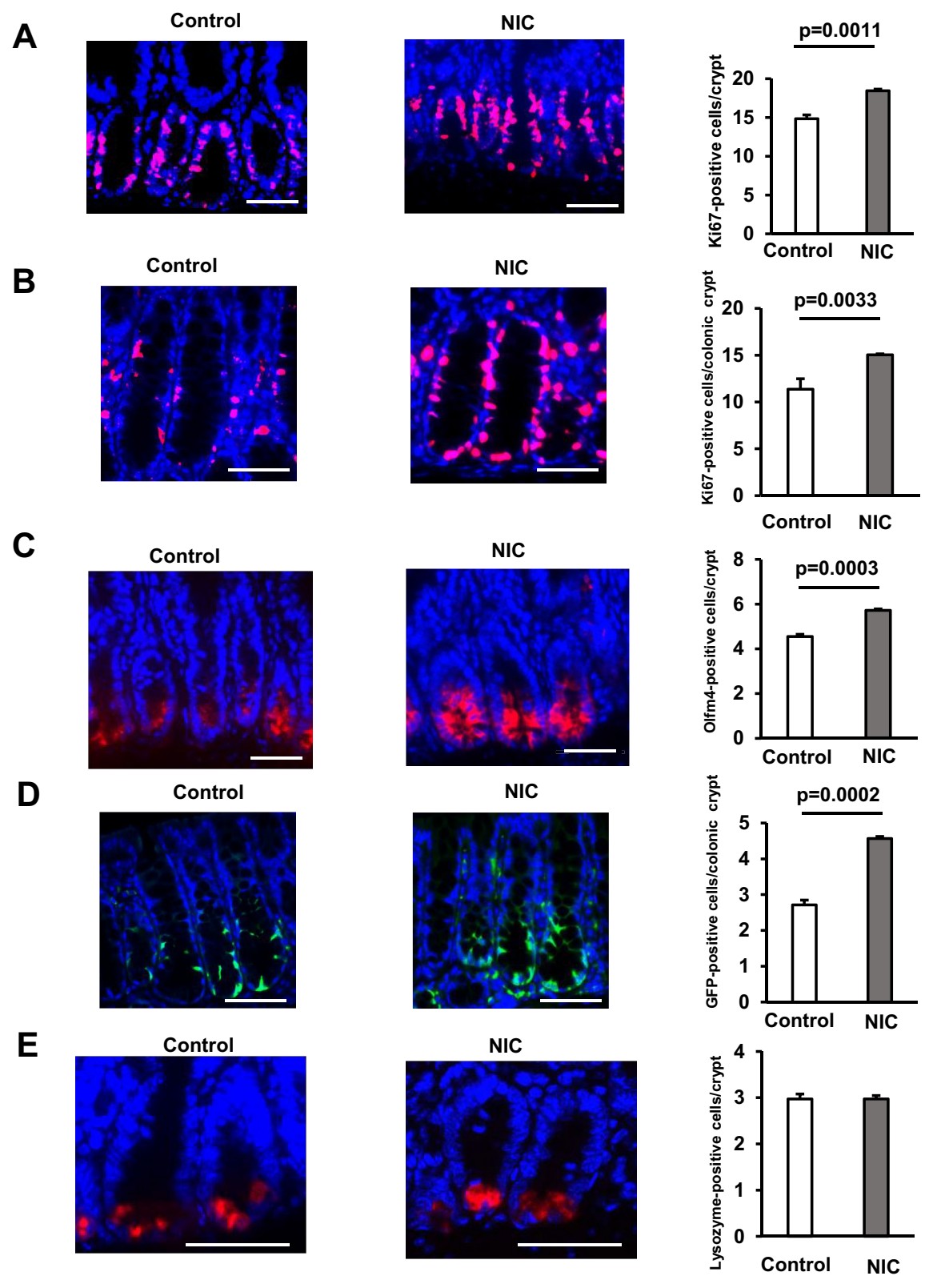

**Figure 1.** Nicotine (NIC) treatment increases the number of ISCs in the Intestine. (**A and B**) Image of Ki67-positive cells (Red: Ki67, Blue: DAPI) and their quantification at the crypt base of proximal jejunum (**A**) or colon (**B**) of NIC-treated and untreated mice (A:3–4 mice per group, B: 3 mice per group). (**C**) Olfm4 staining image (Red: Olfm4, Blue: DAPI) and the quantification of Olfm4-positive cells at the crypt base of proximal jejunum with or without NIC treatment (3–4 mice per group). (**D**) GFP staining image (Green: GFP, Blue: DAPI) and the quantification of LgR5-GFP-positive cells at the crypt

*Figure 1 continued on next page*

*Figure 1 continued*

base of the colon in NIC or control-treated *Lgr5*-EGFP-IRES-CreERT2 mice (3 mice per group). (**E**) Lysozyme staining image (Red: Lysozyme, Blue: DAPI) and the quantification of Lysozyme-positive Paneth cells with or without NIC treatment (3–4 mice per group). Original magnifications: 400× (**A–E**). Scale bar: 50 μm (**A–E**). Values represent the mean ± SEM. Significant differences are denoted by p values (Student's t-test). See also *Figure 1—figure supplement 1*.

The online version of this article includes the following source data and figure supplement(s) for figure 1:

**Source data 1.** The quantification in immunostaining.

**Figure supplement 1.** NIC suppresses the differentiation of ISCs.

**Figure supplement 1—source data 1.** The quantification in staining.

## Results

### NIC treatment enhances the frequency of ISC in the intestine

Histological analyses of the small intestine in C57BL/6 mice treated with 200 μg/ml NIC (which emulates active smoking) revealed that NIC exposure decreased villus length without affecting crypt size (*Figure 1—figure supplement 1A*), which was consistent with NIC-induced decrease in the number of differentiated cells, including absorptive enterocytes (*Figure 1—figure supplement 1B*) or chromogranin A+enteroendocrine cells (*Figure 1—figure supplement 1C*), in the gut.

We investigated the effect of NIC on the population of proliferative cells in the crypts using Ki67 labeling to mark proliferative stem and progenitor cells in the crypts. NIC exposure increased abundance of Ki67-positive cells in the small and large intestine (*Figure 1A and B*). Consistently, the number of proliferative olfactomedin-4 (Olfm4)-positive ISCs significantly increased in the small intestine of NIC-treated mice (*Figure 1C*). Moreover, the number of LgR5 +colonic stem cells (CSCs) was increased in NIC-treated *Lgr5*-EGFP-IRES-CreERT2 mice expressing EGFP under the control of the *LgR5* promoter (*Figure 1D*). Paneth cells support the proliferation of ISCs. However, we did not observe any changes in the number of Paneth cells in NIC-treated mice (*Figure 1E*). These results indicate that the self-renewal of ISCs in NIC-treated mice increased with a reciprocal decrease in the number of differentiated cells.

### NIC treatment enhances the formation of intestinal organoids from ISCs

Our ISC proliferation analysis using intestinal crypts of wild-type mice revealed that a range of nicotine concentrations (100 nM, 1 μM, and 10 μM) promoted the organoid formation from crypts from the small intestine (*Figure 2A*), which was consistent with the in vivo data (*Figure 1A and C*). However, the same dose of cotinine, a minor tobacco alkaloid, and a major metabolite of NIC (*Tan et al., 2021*) did not exhibit similar effects (*Figure 2A*). Furthermore, the addition of 1 μM NIC promoted the organoid formation from colonic crypts (*Figure 2B*). Next, to address how ISCs and Paneth cells interact functionally, we isolated Lgr5-positive ISCs and Paneth cells from control or NIC-treated *Lgr5*-EGFP-IRES-CreERT2 mice as described previously (*Igarashi and Guarente, 2016*). ISCs and Paneth cells were co-cultured in the culture media containing glycogen synthase kinase 3β (GSK3β) inhibitor CHIR99021, which induces β-catenin and thus stimulates organoid formation (*Igarashi and Guarente, 2016*; *Yin et al., 2014*). Lgr5-positive ISCs isolated from NIC mice formed more organoid colonies than those isolated from control mice when cultured with or without Paneth cells (*Figure 2C*). Consistently, the addition of 1 μM NIC to control ISCs stimulated the organoid colony formation (*Figure 2D*). However, Paneth cells exhibited no significant difference in function (organoid formation) between control and NIC-treated groups, with or without CHIR99021 (*Figure 2*; *Figure 2—figure supplement 1*). Therefore, this ex vivo assay for ISC function and ISC–Paneth cell interaction demonstrated that NIC directly stimulates ISC proliferation without affecting the supportive function of Paneth cell for ISC.

### The α7 subunits of nAChR control the effects of NIC on ISC proliferation

We investigated the pathway underlying the NIC-regulated proliferation of ISCs. NIC interacts with nicotinic acetylcholine receptors (nAchRs), which are heterodimers of nine types of α subunits (α2–α10) and three types of β subunits (β2– β4)(*Dani, 2015*). We validated the significance of nAChR signal

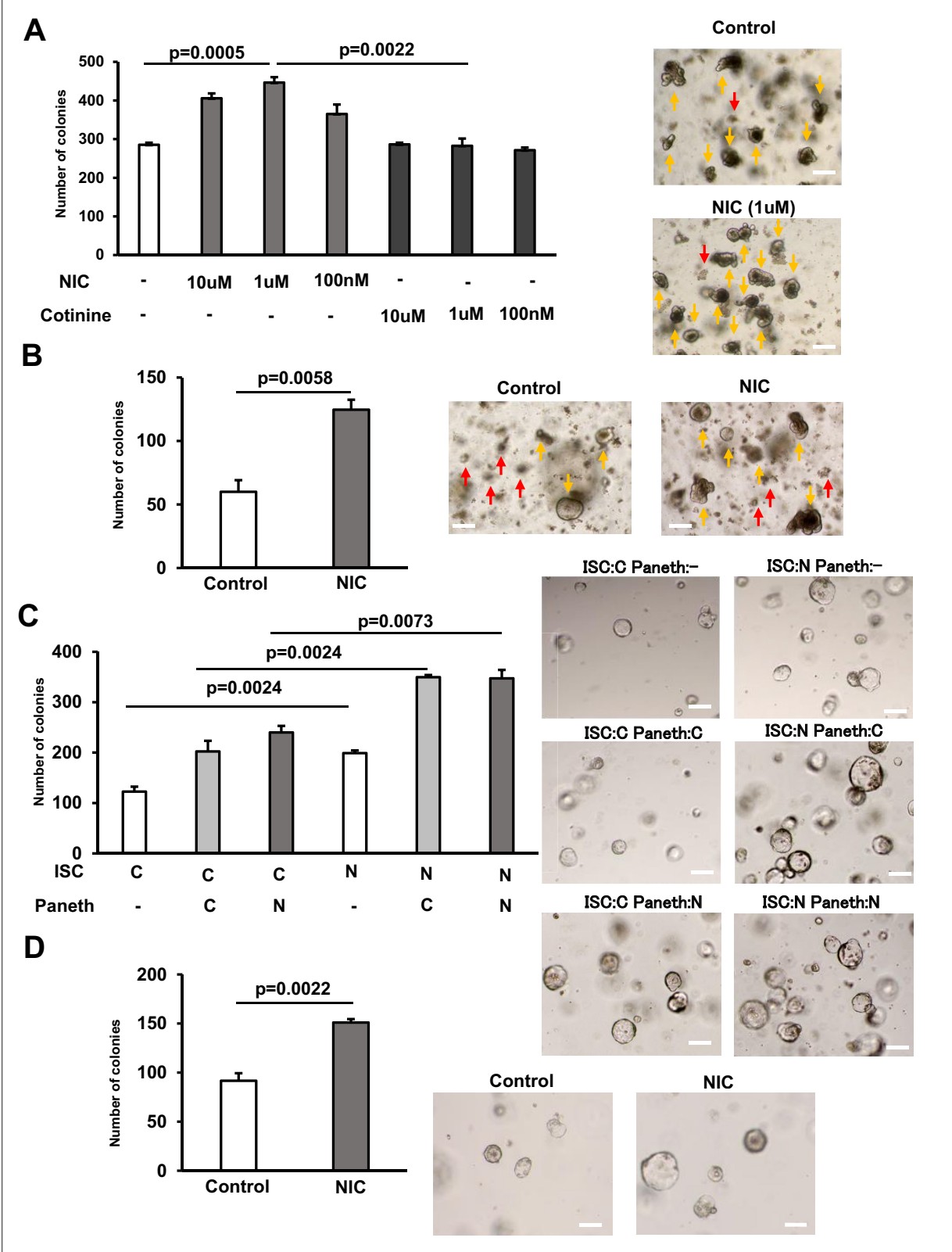

**Figure 2.** NIC enhances the formation of intestinal organoids from ISCs. (**A**) Crypts from the proximal small intestine were cultured with 10 µM, 1 µM, and 100 nM NIC or cotinine to allow ISCs to form organoid colonies; the control set contained no NIC or cotinine. Representative images of the organoids and the quantification of organoids number at day 5 (3 wells/ group) (yellow arrow marks organoids and red arrow indicates aborted crypts). (**B**) Colonic crypts were cultured with or without 1 µM NIC to allow CSCs to form organoid colonies. Representative images of the organoids and the

*Figure 2 continued on next page*

*Figure 2 continued*

quantification of organoids number at day 5 are shown (3 wells/ group) (yellow arrow marks organoids and red arrow indicates aborted crypts). (**C**) ISCs and Paneth cells were isolated from the small intestine of *Lgr5*. *EGFP-IRES-CreERT2* mice treated with or without NIC; 2×10³ cells each were co-cultured in the medium containing 10 μM CHIR99021. Representative images of the organoids and the frequency of organoids at day 5 (3 wells/ group). (**D**) ISCs isolated from the small intestine of *Lgr5-EGFP-IRES-CreERT2* mice were cultured in the absence of Paneth cells using the medium containing 10 μM CHIR99021, with or without 1 μM NIC. Representative images of the organoids and the frequency of organoids number at day 5 (3 wells/ group). C: control, N: NIC. Original magnification: 40×. Scale bar: 100 μm. Values represent the mean ± SEM. Significant differences are denoted by p values (Student's t-test). See also *Figure 3—figure supplement 1*.

The online version of this article includes the following source data and figure supplement(s) for figure 2:

**Source data 1.** The quantification of colonies in organoid assay.

**Figure supplement 1.** NIC induces the formation of intestinal organoids from ISCs.

**Figure supplement 1—source data 1.** The quantification of colonies in organoid assay.

transduction using ISCs, isolated wild-type mice cultured in the presence of NIC, and the nonselective nAChR antagonist Mecamylamine, which indicated that Mecamylamine treatment completely abolishes the NIC-mediated formation of ISC-derived organoids (*Figure 3A*). We further explored the nAChR subtypes. Considering that NIC has a high affinity for the nAChR comprised of α4 and β2 subunits (*McGranahan et al., 2011*), we cultured ISCs in the presence of NIC and Adiphenine, a non-competitive inhibitor of nAChR (α1, α3β4, α4β2, and α4β4). However, Adiphenine exhibited no effect on the NIC-induced organoid formation from ISCs, indicating that the effect of NIC was not mediated by these nAChRs (*Figure 3B*). As some cancer stem cells are known to express α7-nAChR (*Egleton et al., 2008*; *Hirata et al., 2010*), we further analyzed the role of α7-nAChR. The existence of α7-nAChR in ISCs was detected by immunoblotting and RT-PCR, respectively (*Figure 3C and D*). The higher expression of α7 and α9 subunits in both ISCs and Paneth cells were observed compared with other nAChR subunits. However, there were no significant difference in expression of α7 or α9 subunit between ISCs and Paneth cells (*Figure 3C*). To investigate the distribution of nAChRs subunits in human intestine, we analyzed scRNA-seq datasets of the human intestinal epithelium (*Elmentaite et al., 2021*). In consistent with mouse data (*Figure 3C*), the expression of human α7 subunit (CHRNA7) is higher than that of other subunits in both ISCs and Paneth cells (*Figure 3—figure supplement 1*), although the predominance of the expression in ISCs or Paneth cells was not clear (*Figure 3—figure supplement 1*).

Interestingly, NIC treatment significantly upregulated α7-nAChR in ISCs rather than Paneth cells (*Figure 3C and D*). Importantly NIC treatment did not induce other nAChR subunits in ISCs (*Figure 3C*). Moreover, addition of the α7-selective nAChR agonist PNU282987 increased organoid colony formation from ISCs (*Figure 3E*), which was consistent with the effects of NIC. Furthermore, α-Bungarotoxin, an α7-selective nAChR antagonist, completely inhibited the NIC-induced increase in organoid formation (*Figure 3F*). These outcomes indicate the effect of nicotine is mediated via the α7 subunits of nAChR.

## NIC induces a Hippo-YAP/TAZ and Notch signaling in ISCs

As NIC can activate protein kinase C (PKC) or cAMP-dependent protein kinase A (PKA) via α7-nAChR activation (*Hirata et al., 2010*; *Dajas-Bailador et al., 2002*), we cultured isolated ISCs in the presence of NIC combined with Gö 6983, a pan-PKC inhibitor, or H89 dihydrochloride, a selective PKA inhibitor. H89 dihydrochloride did not suppress NIC-induced organoid formation, however, Gö 6983 completely abolished this effect of NIC (*Figure 4A and B*). Additionally, we cultured ISCs with NIC and Sotrastaurin, another pan-PKC inhibitor inactive to PKC ζ , the loss of which is reported to increase ISC activity both in vivo and in vitro (*Llado et al., 2015*). The observed results with Gö 6983 was reproduced by Sotrastaurin treatment (*Figure 4—figure supplement 1A*), demonstrating other PKCs than PKC ζ mediates the effect of NIC. Consistently, the PKC activator Ingenol-3-angelate stimulated the formation of organoid colonies from ISCs (*Figure 5E and F*).

Next, to investigate the possible downstream signaling of α7-nAChR and PKC associated with NIC-induced renewal of ISCs, we explored PI3K/AKT signaling (*Hers et al., 2011*), p38 /mitogen-activated protein kinase (MAPK) signaling (*Rodríguez-Colman et al., 2017*), and mTORC1 signaling (*Igarashi and Guarente, 2016*; *Igarashi et al., 2019*), each of which plays a crucial role in the expansion of ISCs.

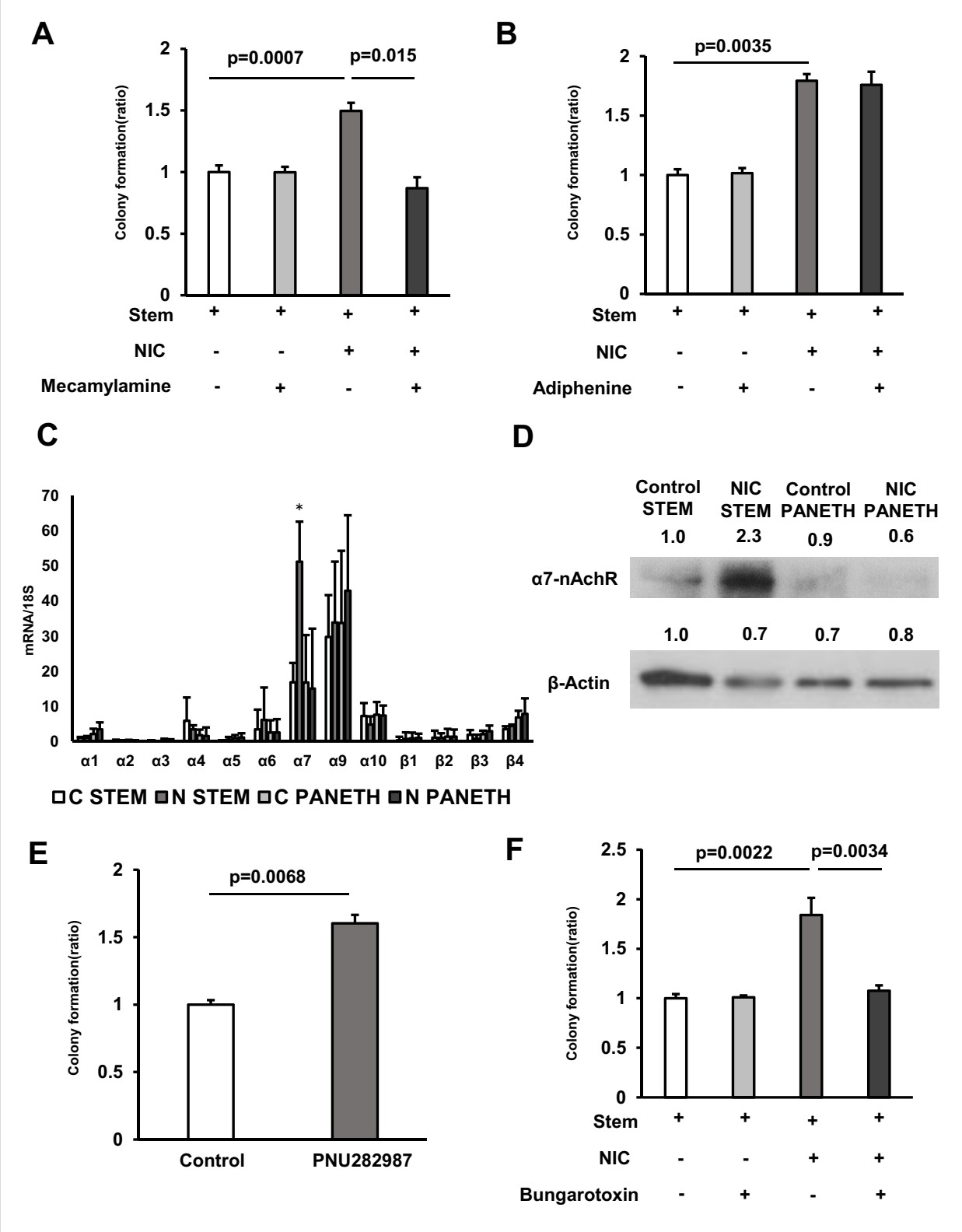

**Figure 3.** The effect of NIC is mediated via the α7 subunits of α7-nicotinic acetylcholine receptor (nAChR). (**A**) Isolated ISCs were cultured in a medium with or without 1 μM NIC and 10 μM Mecamylamine (3 wells/group). (**B**) Isolated ISCs were cultured in a medium with or without 1 μM NIC and 3 μM Adiphenine hydrochloride (3 wells/group). (**C**) In ISCs and Paneth cells isolated from control and NIC mice, mRNA levels of nAChR subunits were analyzed using quantitative real-time PCR (n=5 per group; C: control, N: NIC). ∗$p < 0.05$ (vs C STEM) (Student's t-test). (**D**) ISC or Paneth cell lysates

*Figure 3 continued on next page*

*Figure 3 continued*

prepared from control and NIC mice were immunoblotted with antibodies against α7-nAchR and β-actin. (**E**) Isolated ISCs were cultured in a medium supplemented with or without 10 μM PNU 282987 (3 wells/group). (**F**) Isolated ISCs were cultured in a medium with or without 1 μM NIC and 1 μM α-Bungarotoxin (3 wells/group). Values represent the mean ± SEM. Significant differences are denoted by p values (Student's t-test). See also *Figure 3— figure supplement 1*.

The online version of this article includes the following source data and figure supplement(s) for figure 3:

**Source data 1.** Colony quantification and qRT-PCR.

**Source data 2.** Immunoblotting data with labeling.

**Source data 3.** Immunoblotting raw data.

**Figure supplement 1.** The distribution of human nAchR subunits in ISCs and Pante cells.

Results indicate that NIC does not induce these cascades and these were not required for the effects of NIC on ISCs (*Figure 4—figure supplement 1B–E*).

YAP/TAZ, the downstream effectors of the Hippo signaling pathway, and Notch receptor 1 (Notch1)/ Delta-Like protein 1(Dll1),the main components of Notch signaling are reported to be upregulated in NIC-treated organoids (*Takahashi et al., 2018*). Both the Hippo-YAP/TAZ and Notch signaling pathways regulate intestinal homeostasis via the control of ISC function (*Mo et al., 2014*; *Sancho et al., 2015*) and Notch signaling is positively regulated by YAP/TAZ in the intestine (*Zhou et al., 2011*). Hence, we examined Hippo-YAP/TAZ and Notch signaling. YAP1 and TAZ expression was significantly induced in the crypts of NIC-treated mice (*Figure 4C*). *YAP1* and *TAZ* also upregulated at mRNA level in ISCs from NIC mice (*Figure 4D*). Moreover, activation of Notch signaling in crypts of NIC-treated mice was confirmed through immunoblotting assay (*Figure 4E*). The expression of genes, including *Jagged1*, *HeyL*, *Hes1*, *Hes5*, and *Dll1*, involved in Notch signaling, significantly increased in ISCs obtained from NIC-treated mice (*Figure 4D*). Notably, Hippo-YAP/TAZ and Notch signaling were not significantly activated in Paneth cells, indicating that NIC affects α7-nAchR of ISCs rather than Paneth cells.

Collectively, these results suggest that NIC induces a Hippo-YAP/TAZ and Notch signal pathway in ISCs via activation of α7-nAchR and PKC.

## Inactivation of Hippo-YAP/TAZ and Notch signaling suppresses the NIC-induced colony formation

To further explore the role of the Hippo-YAP/TAZ and Notch signaling in NIC-induced ISC expansion, ISCs were cultured with NIC in the presence of either K-975, a specific inhibitor of transcriptional enhanced associate domain (TEAD), which binds to its transcriptional co-activators YAP or TAZ and forms a transcription complex (*Kaneda et al., 2020*), or γ-secretase inhibitor MK-0752 that inhibits the cleavage of Notch into its active signaling effector, Notch intracellular domain (NICD) (*Krop et al., 2012*). Treatment with K-975 and MK-0752 completely abolished the NIC-induced increased organoid formation (*Figure 5A and B*). Furthermore, K-975 or MK-0752 prevented the increase in the organoid formation in ISCs treated with PNU 298987 or Ingenol-3-angelate (*Figure 5C–F*), suggesting that YAP/Notch signaling acts downstream of α7-nAchR or PKC leading to the response of ISC to NIC (*Figure 5G*).

## DBZ treatment suppresses the expansion of ISCs by NIC in vivo

To validate the significance of Notch signaling in the ISC expansion, NIC-treated mice were subjected to daily IP injection of γ-secretase inhibitor dibenzazepine (DBZ) (1 mg/kg body weight) for 2 weeks (*Figure 6A*). We confirmed that DBZ treatment significantly downregulated Hes5 protein expression in the crypts of NIC-treated mice (*Figure 4B*). Notably, DBZ suppressed the expression of YAP and TAZ in NIC-treated mice (*Figure 6B*). In the intestine, YAP/TAZ regulates Notch signaling (*Zhou et al., 2011*) and Notch activation can activate YAP/TAZ (*Totaro et al., 2018*). Consistent with previous reports, our results demonstrate that Notch inhibitor suppresses YAP/TAZ and Notch activities, indicating a positive feedback loop between Hippo-YAP/TAZ and Notch signaling in ISCs of NIC mice (*Figure 5G*).

Ki67 labeling and immunostaining for the ISC marker Olfm4 elucidated the effect of DBZ treatment on the frequency of ISCs in control and NIC-treated mice. DBZ treatment did not alter the population

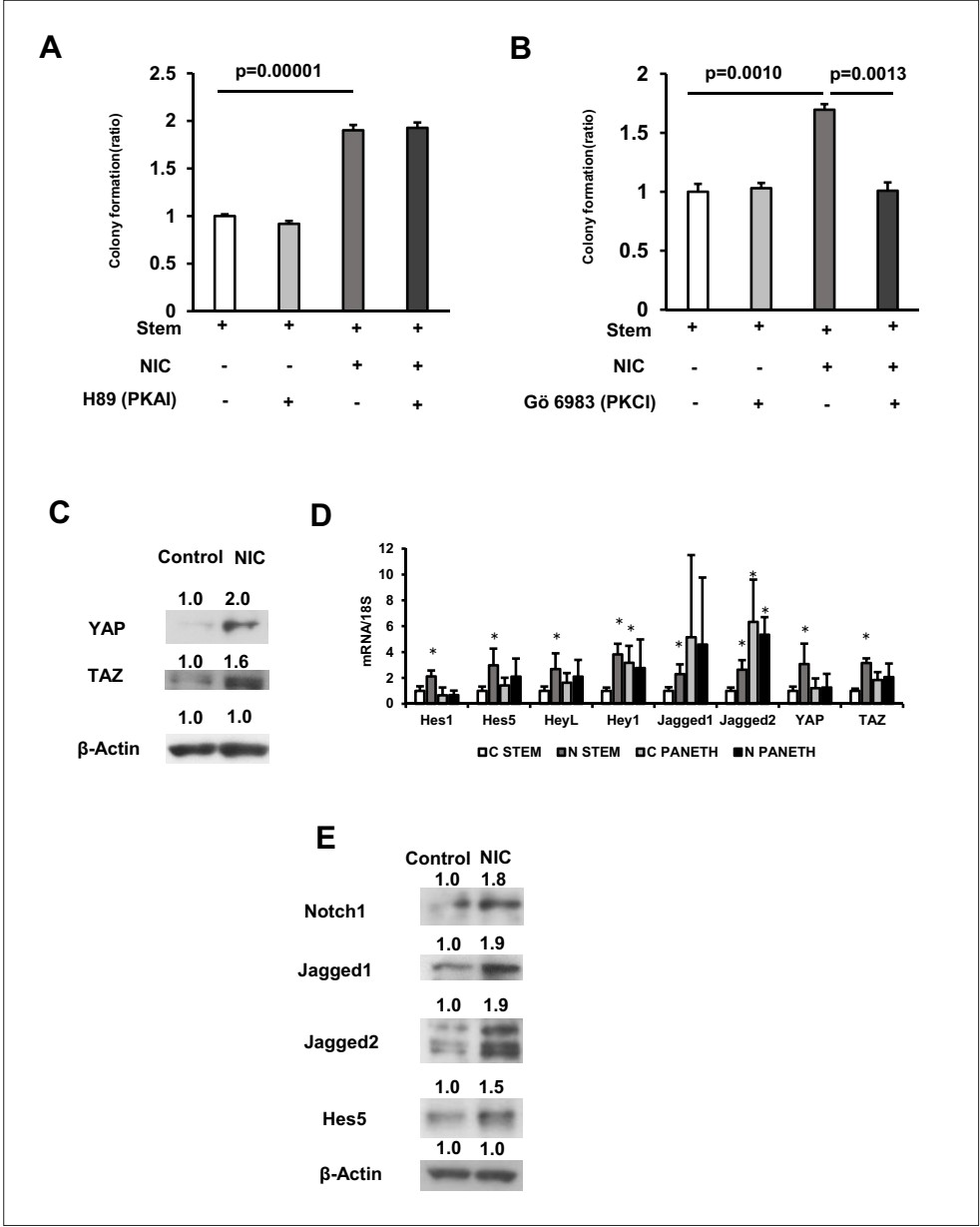

**Figure 4.** NIC induces Hippo-YAP/TAZ and notch signaling in ISCs. (**A and B**) Isolated ISCs cultured using a medium supplemented with or without 1 μM NIC combined with either (**A**) 1 μM H89 dihydrochloride (PKA inhibitor) or (**B**) 10 nM Gö 6983 (PKC inhibitor; 3 wells/group). (**C**) Crypt lysates isolated from control and NIC-treated mice were immunoblotted using antibodies against YAP, TAZ, and β-actin. (**D**) In ISCs or Paneth cells (n=5 per group) isolated from control or NIC mice, mRNA levels of genes associated with Hippo-YAP/TAZ and Notch signaling were determined through quantitative real-time PCR. ∗$p < 0.05$ (vs C STEM) (Student's t-test). (**E**) Crypt lysates obtained from control and NIC-treated mice were immunoblotted using antibodies against Notch1, Jagged1, Jagged2, Hes5, and β-actin. Values represent the mean ± SEM. Significant differences are denoted by p values (Student's t-test). See also *Figure 4—figure supplement 1*.

The online version of this article includes the following source data and figure supplement(s) for figure 4:

**Source data 1.** Colony quantification and qRT-PCR.

**Source data 2.** Immunoblotting data with labeling.

**Source data 3.** Immunoblotting raw data.

**Figure supplement 1.** PI3K/AKT, mTORC1, and p38 /MAPK signaling cascades do not mediate the effect of NIC.

**Figure supplement 1—source data 1.** The quantification of colonies in organoid assay.

*Figure 4 continued on next page*

*Figure 4 continued*

**Figure supplement 1—source data 2.** Immunoblotting data with labeling.

**Figure supplement 1—source data 3.** Immunoblotting raw data.

of ISCs in the control mice but significantly suppressed the expansion of Ki67 +and Olfm4 + cells in NIC-treated mice (*Figure 6C and D*). Moreover, DBZ suppressed the expansion of Ki67 + cells and CSCs in the colon of NIC-treated mice (*Figure 6—figure supplement 1*). These outcomes demonstrate that the Hippo-YAP/TAZ and Notch signaling pathways are crucial for ISC expansion in NIC-treated mice analyzed in vivo.

## DBZ inhibits intestinal tumor growth by NIC

As ISCs are the potential origin of tumors (*Barker et al., 2009*), we hypothesized that NIC-induced ISC expansion can promote tumor formation in a tumor-initiating background, such as Apc loss. We crossed stem cell-specific $Lgr5^{-EGFP-IRES-creERT2}$ knock-in mice with $Apc^{flox/flox}$ mice ($Lgr5^{CreERT2} Apc^{fl/fl}$ mice); in the resulting mice, the transformation of Lgr5–GFP positive stem cells efficiently drives adenoma formation throughout the intestine after Apc loss induced by Cre activation using tamoxifen (*Figure 7A*).

To test whether NIC promotes tumor formation via ISC expansion, tumor formation was induced in NIC-treated $Lgr5^{CreERT2} Apc^{fl/f}$ (NIC- $Lgr5^{CreERT2} Apc^{fl/fl}$) mice, and entire intestines, isolated from them, were examined for polyps (*Figure 7A*). A marked increase in the abundance of polyps was detected throughout the intestine of NIC-treated $Lgr5^{CreERT2} Apc^{fl/fl}$ mice; moreover, these polyps were significantly larger than that in the control mice (*Figure 7B*). Consistently, the area of β-catenin-positive adenomatous lesions throughout the entire intestine significantly increased in NIC- $Lgr5^{CreERT2} Apc^{fl/fl}$ mice (*Figure 7C*). These results indicated that NIC increased the overall polyp burden in the intestines of $Lgr5^{CreERT2} Apc^{fl/fl}$ mice by increasing their size and number.

Finally, to validate whether DBZ can suppress NIC-induced intestinal adenomas, we treated NIC- $Lgr5^{CreERT2} Apc^{fl/fl}$ mice with DBZ 4 weeks after the induction of Apc loss (*Figure 7D*). Notably, this treatment effectively reduced the abundance of polyps and β-catenin positive adenomatous lesions in NIC- $Lgr5^{CreERT2} Apc^{fl/fl}$, indicating the efficiency of DBZ for the prevention and treatment of NIC-induced intestinal tumors (*Figure 7E and F*).

## Discussion

Our data propose a model in which NIC enhances the self-renewal of ISCs via activated Hippo-YAP/TAZ and Notch signaling (*Figure 5G*). Our analyses revealed an increase in the number of ISCs in NIC-treated mice. In contrast to ISCs, Paneth cells showed no increase in cell numbers in NIC-treated mice. Similarly, we assessed the functional ability of ISCs and Paneth cells in terms of ex vivo organoid formation, which indicated a NIC-induced gain-of-function in ISCs but not in Paneth cells. Hence, we propose that the increase of ISC activity induced by NIC are not dependent on Paneth cells in functional assays. Furthermore, we traced the signaling pathway in ISCs and detected that ISCs respond to NIC via α7-nAchR, PKC activation, and stimulation of Hippo-YAP/TAZ and Notch signaling (*Figure 5*). Consistent with this model, DBZ treatment inhibited Hippo-YAP/TAZ and Notch signaling in mice and prevented NIC-induced ISC expansion and transformation through Apc loss.

Treatment of organoids with NIC has been reported to enhance cell growth and expression of marker genes of stem cells (*Takahashi et al., 2018*). Moreover, mRNA analysis of NIC-treated organoids showed that the expression of *YAP1/TAZ* and *Notch1/Dll1* was upregulated after treatment with NIC (*Takahashi et al., 2020*). This previous report predicted that NIC activates the Hippo and Notch signaling pathway in Paneth cells rather than ISCs, because the α2β4-nAChR was mainly expressed in Paneth cells (*Takahashi et al., 2020*). However, the hypothesis was not fully testified in the report. In contrast, our analysis validates that NIC affects ISC rather than Panth cells via Hippo-YAP/TAZ and Notch signaling activated though nAchRa7 by mRNA and protein analyses of sorted ISCs or Paneth cells and ex vivo functional assays.

The crosstalk between Hippo, Notch, and Wnt signaling regulates mammalian intestinal homeostasis (*Totaro et al., 2018*; *Khoramjoo et al., 2022*; *Li et al., 2019*). Activated YAP/TAZ, which act downstream of Hippo signaling, translocate to the nucleus and induce the gene expression of Notch

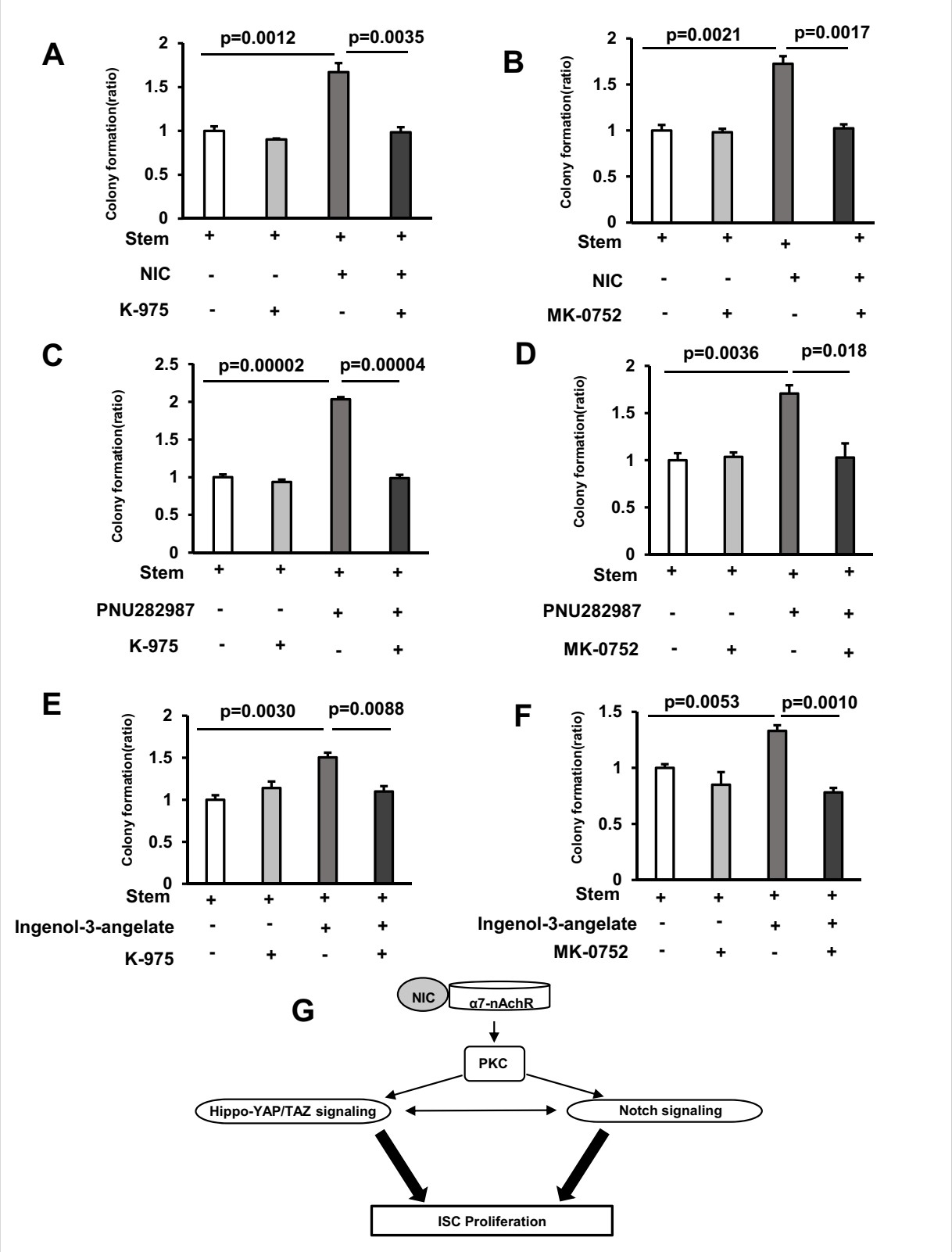

**Figure 5.** Inactivation of Hippo-YAP/TAZ and Notch signaling suppresses NIC-induced Colony Formation in mice. (**A**) Isolated ISCs were cultured using a medium with or without 1 μM Nicotine and 5 nM K-975 (3 wells/group). (**B**) Isolated ISCs were cultured using a medium with or without 1 μM Nicotine and 1 μM MK-0752 (3 wells/group). (**C**) Isolated ISCs were cultured using a medium with or without 10 μM PNU282987 and 5 nM K-975 (3 wells/group). (**D**) Isolated ISCs were cultured using a medium with or without 10 μM PNU282987 and 1 μM MK-0752 (3 wells/group). (**E**) Isolated ISCs were cultured

*Figure 5 continued on next page*

*Figure 5 continued*

in a medium with or without 1 nM Ingenol-3-angelate and 5 nM K-975 (3 wells/group). (**F**) Isolated ISCs were cultured in a medium with or without 1 nM Ingenol-3-angelate and 1 μM MK-0752 (3 wells/group). (**G**) Schematic model of NIC-associated signaling pathway in ISCs. The model traces a signaling cascade via α7-nAChR, PKC, Hippo-YAP/TAZ and Notch signaling in ISCs. Values represent the mean ± SEM. Significant differences are denoted by p values (Student's t-test).

The online version of this article includes the following source data for figure 5:

**Source data 1.** The quantification of colonies in organoid assay.

receptors and Notch ligands, and thus, Notch is a mediator of YAP1-induced ISC expansion (*Zhou et al., 2011*; *Totaro et al., 2018*). Moreover, YAP/TAZ and Notch signaling congruently induces nuclear translocation of YAP/TAZ and NICD, regulating the expression of common target genes (*Totaro et al., 2018*). Furthermore, YAP with TEADs enhances Jagged1-Notch1 signaling; Notch1 promotes YAP stability through inhibited β-TrCP-mediated degradation and formation of a YAP1- jagged-1/Notch1-positive feedback loop in breast cancer cells (*Zhao et al., 2022*). Similarly, we propose a YAP/TAZ-Notch positive feedback loop in ISCs of NIC-treated mice (*Figure 5G*). This model is consistent with DBZ-induced suppression of Hippo-YAP/TAZ and Notch signaling in ISCs from NIC-treated mice (*Figure 6B*). As NIC effectively expands the ISC population through the upregulated α7-nAChR and the formation of the YAP/TAZ-Notch loop, we speculate these factors to be good therapeutic targets for treating NIC-induced colon cancer.

In the intestine, YAP activation enhances the expression of both β-catenin and transcriptional targets of Wnt signaling (*Deng et al., 2018*). Moreover, Notch signaling concomitantly regulates intestinal cell proliferation via Wnt signaling (*Fre et al., 2009*). NIC is supposed to activate Wnt signaling via Hippo-YAP/TAZ and Notch signaling. Similarly, the expression of target proteins (Sox9, TCF4 and, C-myc) of the Wnt/β-catenin pathway as well as cell-cycle regulatory proteins, such as cyclin B, and cyclin E, was up-regulated in crypts obtained from NIC-treated mice in our experiment (data not shown). However, the effect of NIC on ISC organoid formation appears to be independent of CHIR99021, a Wnt activator (*Figure 2*). Moreover, In the *Lgr5*^CreERT2^ *Apc*^fl/fl^ mouse model, APC loss results in a constitutive stabilization of β-catenin, thus the hyperproliferation of ISCs by NIC treatment in this mouse model is likely beyond Wnt activation (*Figure 7*). The downstream pathway of Hippo-YAP/TAZ and Notch signaling remains to be determined.

Our study demonstrated that NIC increased the expression of YAP/TAZ and Notch target genes, as well as the ISC population. Our ex vivo organoid assay revealed the role of Hippo-YAP/TAZ and Notch signaling in NIC-induced ISC expansion. However, the mechanism underlying NIC-activated Hippo-YAP/TAZ and Notch signaling also remains unclear. NIC administration can induce the nuclear translocation and activation of YAP in Esophageal Squamous Cell Cancer (*Zhao et al., 2014*). The nAChRs physically interact with YAP, leading to the upregulation and nuclear translocation of YAP1. This process is considered to be mediated by PKC activation because PKC-specific inhibitors block NIC-induced YAP activation (*Zhao et al., 2014*). Moreover, PKC activity induces the translocation of ADAM-10, which is implicated in the cleavage of Notch receptors (*Bozkulak and Weinmaster, 2009*), to the cell membrane of glioblastoma (*Kohutek et al., 2009*). Hence, we propose the possibility of NIC-induced activation of YAP and ADAM via PKC, leading to Hippo-YAP/TAZ and Notch activation and ISC expansion.

The lifetime risk of various cancers is strongly correlated with frequency of stem cell divisions that maintain tissue homeostasis (*Tomasetti and Vogelstein, 2015*). However, the risk of cancer associated with stem cell divisions is heavily influenced by both extrinsic and intrinsic factors (*Wu et al., 2016*). Extrinsic factors, such as NIC, may increase the risk of cancer by promoting stem cell division. Our data indicated that NIC increased the abundance and proliferation of ISCs, which may partly explain the increased rate of intestinal tumors in smokers.

In conclusion, we demonstrated that NIC enhances the ISC population via α7-nAChR as well as Hippo-YAP/TAZ and Notch signaling. These findings revealed the pivotal role of NIC as a stimulant of the cancer stem cell proliferation in intestinal tumors, and thus, explains colon cancer development in cigarette smokers. The development of drugs that can block the α7-nAChR, Hippo-YAP/TAZ, and Notch signaling may provide a new therapeutic strategy for treating colon cancers.

# Materials and methods

**Key resources table**

| Reagent type (species) or resource | Designation | Source or reference | Identifiers | Additional information |
|---|---|---|---|---|
| Strain, strain background (*Mus musculus*) | *Lgr5-EGFP-IRES-CreERT2* mice | Jackson Laboratory | #008875 | |
| Strain, strain background (*Mus musculus*) | *Apc CKO* mice | National Cancer Institute | #01XAA | |
| Strain, strain background (*Mus musculus*) | *Rosa26-CAG-lsl-tdTomato* mice | Jackson Laboratory | #007909 | |
| Antibody | rabbit monoclonal anti-Ki67 | Cell Signaling Technology | #12202 | IHC (1:200) |
| Antibody | rabbit monoclonal anti-Olfm4 | Cell Signaling Technology | #39141 | IHC (1:400) |
| Antibody | mouse monoclonal anti-GFP | Santa Cruz Biotechnology | #sc-9996 | IHC (1:50) |
| Antibody | rabbit polyclonal anti-Lysozyme | Thermo Fisher Schientific | #PA5-16668 | IHC (1:50) |
| Antibody | mouse monoclonal anti-chromogranin A | Santa Cruz Biotechnology | #sc-393941 | IHC (1:50) |
| Antibody | rabbit polyclonal anti-β-catenin | Cell Signaling Technology | #9562 | IHC (1:200) |
| Antibody | goat polyclonal anti-mouse IgG H&L (Alexa Fluor 488) | Abcam | #ab150113 | IHC (1:200) |
| Antibody | mouse monoclonal anti-β-actin | Santa Cruz | #sc-47778 | WB (1:200) |
| Antibody | mouse monoclonal anti-YAP | Santa Cruz | #sc-101199 | WB (1:200) |
| Antibody | mouse monoclonal anti-TAZ | Santa Cruz | #sc-293183 | WB (1:200) |
| Antibody | mouse monoclonal anti-α7-AchR | Santa Cruz | #sc-58607 | WB (1:200) |
| Antibody | mouse monoclonal anti-Notch1 | Santa Cruz | #sc-376403 | WB (1:100) |
| Antibody | mouse monoclonal anti-Jagged1 | Santa Cruz | #sc-390177 | WB (1:100) |
| Antibody | mouse monoclonal anti-Jagged2 | Santa Cruz | #sc-515725 | WB (1:100) |
| Antibody | mouse monoclonal anti-Hes5 | Santa Cruz | #sc-293445 | WB (1:200) |
| Antibody | mouse monoclonal anti-p38 | Santa Cruz | #sc-81621 | WB (1:200) |
| Antibody | mouse monoclonal anti-phospho-p38 | Santa Cruz | #sc-166182 | WB (1:100) |
| Antibody | rabbit monoclonal anti-S6 | Cell Signaling | #2217 | WB (1:1000) |
| Antibody | rabbit monoclonal anti-phospho-S6 Ser235/236 | Cell Signaling | #4858 | WB (1:2000) |
| Antibody | anti-Mouse IgG, HRP-Linked Whole Ab Sheep | Cytiva | NA931 | WB (1:5000) |
| Antibody | anti-Rabbit IgG, HRP-Linked Whole Ab Donkey | Cytiva | NA934 | WB (1:5000) |
| Antibody | rat monoclonal APC-conjugated anti-mouse CD24 Antibody | Biolegend | #101814 | FCY (1:500) |
| Commercial assay or kit | TSA Plus Cyanine 3 System | Akoya Biosciences | #NEL744001KT | |
| Chemical compound, drug | Mecamylamine | Cayman Chemical | #14602 | |
| Chemical compound, drug | Adiphenine hydrochloride | MedChemExpress | #HY-B0379A | |
| Chemical compound, drug | PNU282987 | MedChemExpress and Cayman Chemical | #17424 #HY-12560A | |
| Chemical compound, drug | α-Bungarotoxin | R&D | #2133/1 | |
| Chemical compound, drug | H-89 dihydrochloride | MedChemExpress | #HY-15979A | |
| Chemical compound, drug | Gö 6983 | Cayman Chemical | #13311 | |

*Continued on next page*

*Continued*

| Reagent type (species) or resource | Designation | Source or reference | Identifiers | Additional information |
|---|---|---|---|---|
| Chemical compound, drug | Sotrastaurin | MedChemExpress | #HY-10343 | |
| Chemical compound, drug | K-975 | MedChemExpress | #HY-138565 | |
| Chemical compound, drug | MK-0752 | MedChemExpress | #HY-10974 | |
| Chemical compound, drug | Ingenol-3-angelate | Cayman Chemical | #16207 | |
| Chemical compound, drug | AKT Inhibitor VIII | MedChemExpress | #HY-10355 | |
| Chemical compound, drug | SB203580 | Tokyo Chemical Industry | #F0864 | |
| Chemical compound, drug | Rapamycin | LKT Laboratories, Inc | #R0161 | |
| Chemical compound, drug | Collagenase Type IV | Worthington Biochemical Corporation | #CLS4 | |
| Chemical compound, drug | Valproic acid sodium salt | FUJIFILM Wako Pure Chemical Corporation | #2815/100 | |
| Chemical compound, drug | [-]-Cotinine | Sigma-Aldrich | #C-016 | |
| Chemical compound, drug | Nicotine hemisulfate salt | Sigma-Aldrich | #N1019 | |
| Chemical compound, drug | DBZ (Dibenzazepine) | Cayman Chemical | #14627 | |
| Chemical compound, drug | Tamoxifen | Cayman Chemical | #13258 | |
| Chemical compound, drug | Mounting medium With DAPI Aqueous Fluoroshield | Abcam | #ab104139 | |
| Chemical compound, drug | Can Get Signal Immunoreaction Enhancer Solution | TOYOBO | NKB-101 | |

## Animals

*Lgr5*[-EGFP-IRES-CreERT2] mice were purchased from Jackson Laboratory (Bar Harbor, ME). *Apc*[flox] mice were obtained from the National Cancer Institute. *Lgr5*[-EGFP-IRES-CreERT2] mice were crossed with *Apc*[flox/flox] mice (*Lgr5*[CreERT2] *Apc*[fl/fl]). *Lgr5*[CreERT2] *Apc*[fl/fl]:tdTomato mice were generated by crossing *Lgr5*[CreERT2] *Apc*[fl/fl] mice with *Rosa26-CAG-lsl-tdTomato* mice from Jackson Laboratories. All lines were maintained in the C57BL/6 background.

Mice were housed in a controlled environment maintaining 12 hr:12 hr light:dark cycle at 25 ± 1 °C. This study was performed following the guidelines of the Animal Care Committee of the University of Tokyo. All of the animals were handled according to the protocol approved by the Committee on the Ethics of Animal Experiments of the University of Tokyo (Permit Number: A2023M043-04).

## Nicotine treatment

Mice were administered with 200 µg/ml NIC (Nicotine hemisulfate salt, Sigma-Aldrich, Saint Louis, MO) through drinking water for more than 8 weeks. Water bottles were replaced every alternate day.

## Tamoxifen treatment

Recombination by *Lgr5*[CreERT2] in *Lgr5*[CreERT2] *Apc*[fl/fl]:tdTomato mice was induced with a single dose of tamoxifen (30 mg kg$^{-1}$ body weight) suspended in corn oil, administered through the intraperitoneal injection. *Lgr5*[CreERT2]-induced mice were analyzed 4 weeks after induction.

## DBZ treatment

The mice were intraperitoneally injected with DBZ (1 mg/kg body weight; Cayman Chemical, Ann Arbor, MI) or DMSO (in PBS) for 2 or 4 weeks.

## Crypt isolation and culture

Crypts were isolated as described previously (*Igarashi and Guarente, 2016*; *Igarashi et al., 2019*). For this purpose, the proximal half of the small intestine was isolated, opened longitudinally, and

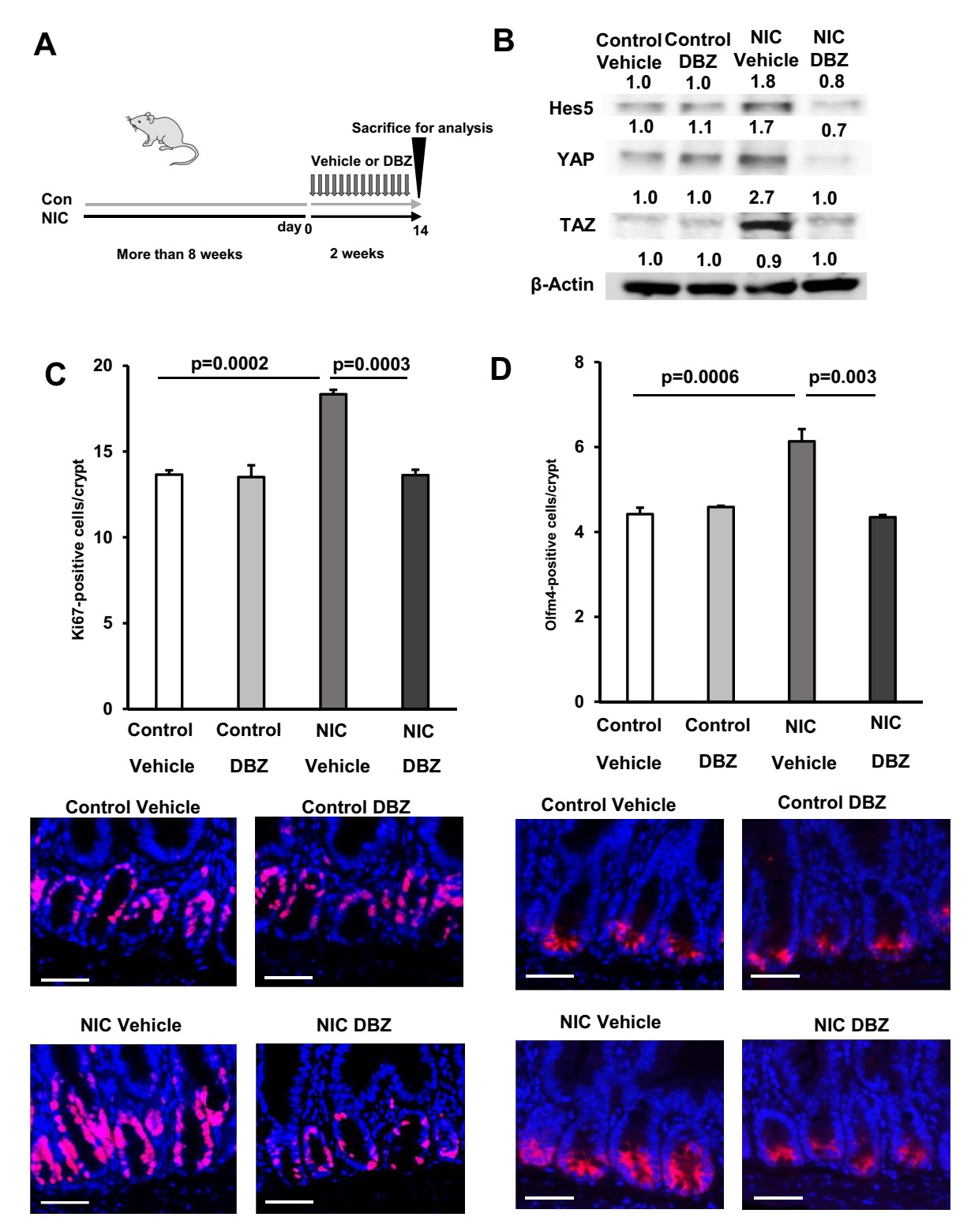

**Figure 6.** Dibenzazepine (DBZ) treatment suppresses the NIC-induced expansion of ISCs in vivo. (**A**) Schematic representation of the treatment showing daily injection of DBZ (1 mg/kg body weight) for 2 weeks. (**B**) Immunoblotting analysis of crypt lysate isolated from DBZ- and vehicle-treated mice in control and NIC-treatment groups using Hes5, YAP, TAZ, and β-actin antibodies. (**C and D**) Immunostained Ki67-positive and (**C**) (Red, Ki67; Blue, DAPI) Olfm4-positive cells (**D**) (Red, Olfm4; blue, DAPI) and their quantification in the proximal jejunum of DBZ- or vehicle-treated mice (NIC-treated

*Figure 6 continued on next page*

*Figure 6 continued*

and untreated) (n=3 per group). Original magnifications: ×400 (**C and D**). Scale bar: 50 µm (**C and D**). Values represent the mean ± SEM. Significant differences are denoted by p values (Student's t-test). See also *Figure 6—figure supplement 1*.

The online version of this article includes the following source data and figure supplement(s) for figure 6:

**Source data 1.** The quantification in immunostaining.

**Source data 2.** Immunoblotting data with labeling.

**Source data 3.** Immunoblotting raw data.

**Figure supplement 1.** DBZ treatment suppresses the NIC-induced expansion of CSCs in vivo.

**Figure supplement 1—source data 1.** The quantification in immunostaining.

washed with cold PBS. After washing the intestine with cold PBS, it was cut into small (5 mm long) pieces with scissors and washed using cold PBS. Subsequently, the pieces were gently incubated in PBS supplemented with 5 mM EDTA for 40 min at 4 °C and resuspended in ice-cold PBS without EDTA followed by vigorous shaking performed manually for crypt isolation. Isolated crypts were filtered using a 70 µm mesh (Corning), collected in crypt culture medium, quantified, and embedded in Matrigel (Corning, Inc Corning, NY). In a 48-well plate, 300 isolated crypts were plated per well and cultured using crypt culture medium, Dulbecco's Modified Eagle Medium:Nutrient Mixture F-12 (DMEM/F12) (Thermo Fisher Scientific) supplemented with 1×N2 (Thermo Fisher Scientific), 1×B27 (Thermo Fisher Scientific), 1 mM N-acetyl-L-cysteine (Sigma-Aldrich), 50 ng/mL EGF (PeproTech, Inc, Cranbury, NJ), 100 ng/mL Noggin (PeproTech, Inc), and R-spondin1 conditioned media (R&D Systems, Minneapolis, MN). The medium was changed in 2 days. Nicotine hemisulphate salt or cotinine ([-]-Cotinine, Sigma-Aldrich) was added to the crypt culture medium as required for the specific analyses. The number of alive organoids was counted under microscope 5 days after plating.

## Flow cytometry

ISC and Paneth cells were isolated from dissociated intestinal crypts using flow cytometry as described previously (*Igarashi and Guarente, 2016*; *Igarashi et al., 2019*). The crypts were centrifuged for 5 min at 300×$g$ at 4 °C and the pellets were gently resuspended in 800 µl TrypLE Express (Thermo Fisher Scientific) supplemented with 200 µl PBS followed by incubation in a water bath at 32 °C for 1.5 min; after incubation, the samples were placed on ice. Next, 12 mL of cold minimum essential medium (MEM; FUJIFILM Wako Pure Chemical Corporation, Osaka, Japan) was added, and the samples were gently triturated twice. After centrifugation for 5 min at 200×$g$ at 4 °C, the pellets were resuspended and incubated for 15 min on ice in 0.5 ml MEM containing CD24-APC antibody (1:500, 101814, Biolegend, San Diego, CA). After centrifugation for 5 min at 200×$g$ at 4 °C, the pellets were resuspended in MEM containing 1.5 µM propidium iodide (PI) (Fujifilm Wako Pure Chemical Corporation). The samples were filtered through a 40 µm mesh (Corning) and immediately sorted using a BD FACS Aria III Cell Sorter (BD Life Sciences, San Jose, CA). ISCs were isolated as Lgr5-EGFP$^{hi}$CD24$^{low}$PI$^-$ and Paneth cells were isolated as CD24$^{hi}$SideScatter$^{hi}$Lgr5-EGFP$^-$PI$^-$.

## Co-culture of isolated ISCs and Paneth cells

Isolated ISCs and Paneth cells were suspended separately in the medium containing 1×N2, 1×B27, and 10 µM Y-27632 (FUJIFILM Wako Pure Chemical Corporation). ISCs (2000 cells) and Paneth cells (2000 cells) were then seeded into 30 µl Matrigel containing 1 µM Jagged-1 (AnaSpec, San Jose, CA) and 10 µM Y-27632. The matrigel drops with ISCs and Paneth cells were allowed to solidify on a 48-well plate for 15 minutes in a 37 °C incubator. The culture medium containing 1×N2, 1×B27, 1 mM N-Acetyl-cysteine, 50 ng/ml EGF, 200 ng/ml Noggin, R-spondin1 conditioned media (R&D Systems, Minneapolis, MN), and CHIR99021 (FUJIFILM Wako Pure Chemical Corporation) was then added onto the drops of matrigel followed by incubation at 37 °C incubator. Isolated ISCs and Paneth Cells were co-cultured without CHIR99021 (*Figure 2—figure supplement 1A*).

Other supplements, including Nicotine hemisulfate salt, Mecamylamine (Cayman Chemical), Adiphenine hydrochloride (MedChemExpress), PNU282987(MedChemExpress and Cayman Chemical), α-Bungarotoxin (R&D), H-89 dihydrochloride (MedChemExpress), Gö 6983 (Cayman Chemical), Sotrastaurin (MedChemExpress), K-975 (MedChemExpress), MK-0752 (MedChemExpress), Ingenol-3-angelate (Cayman Chemical), AKT Inhibitor VIII (MedChemExpress), SB203580 (Tokyo Chemical

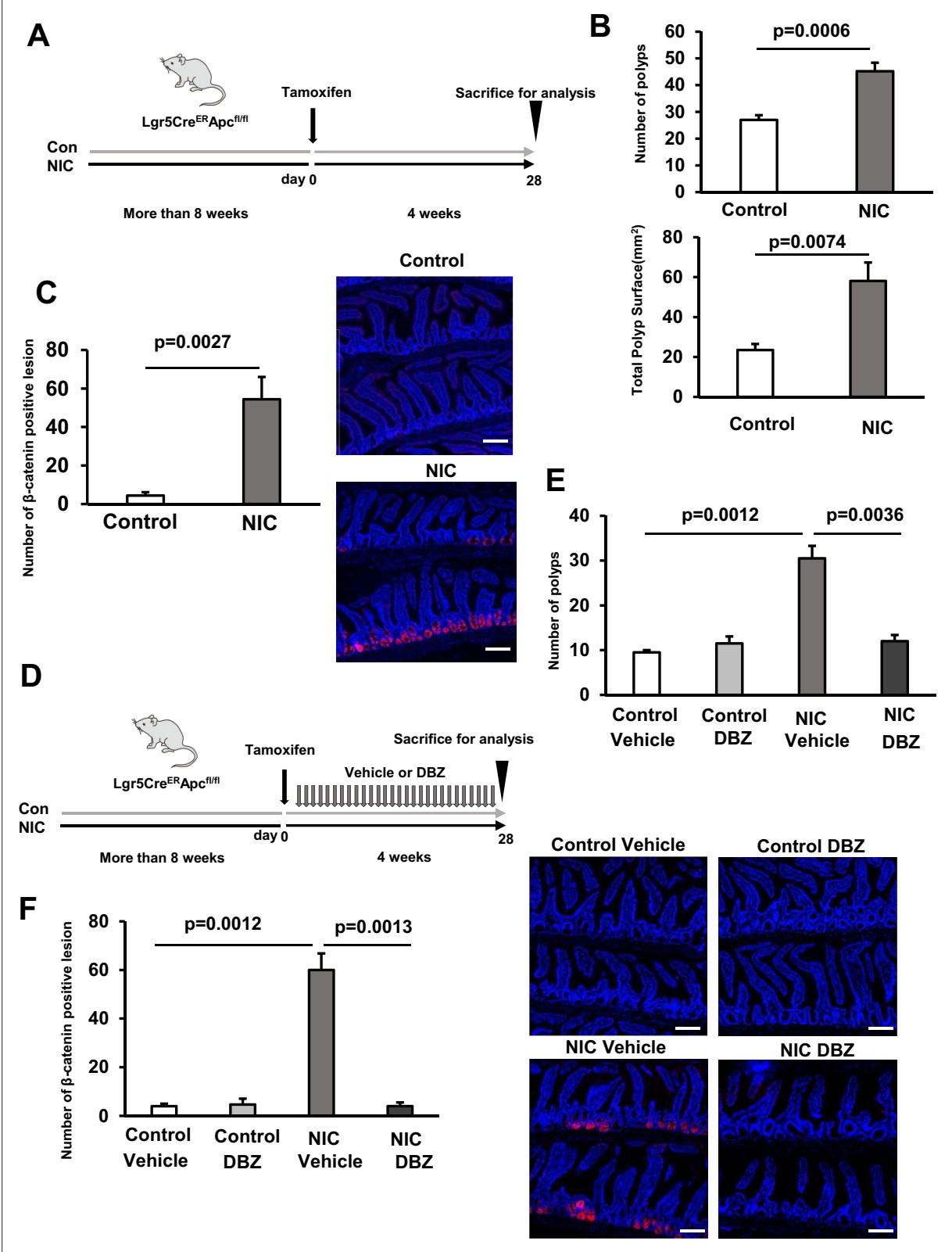

**Figure 7.** BZ inhibits intestinal tumor growth by NIC. (**A**) Schematic representation of *Apc*<sup>flox/flox</sup>; *Lgr5*<sup>-EGFP-IRES-CreERT2</sup> (*Lgr5*<sup>CreERT2</sup> *Apc*<sup>fl/fl</sup>) tumor initiation. Mice were treated with control or NIC more than 8 weeks before a single Tamoxifen injection (30 mg/kg body weight), continued for 4 weeks before tissue collection. (**B**) Macroscopic quantification of the number and area of polyps in the entire intestine of control or NIC-treated *Lgr5*<sup>CreERT2</sup> *Apc*<sup>fl/fl</sup> mice. (**C**) Representative images (Red: β-catenin, Blue: DAPI) and the quantification of the number of β-catenin positive adenomatous lesions in the

*Figure 7 continued on next page*

*Figure 7 continued*

entire intestine of control or NIC-treated *Lgr5^CreERT2 Apc^fl/fl* mice. (**D**) Schematic presentation of *Lgr5^CreERT2 Apc^fl/fl* tumor initiation. Control or NIC-treated *Lgr5^CreERT2 Apc^fl/fl* mice were subjected to a single Tamoxifen injection (30 mg/kg body weight), followed by daily DBZ or vehicle injections continued for 4 weeks before tissue collection. (**E**) Macroscopic quantification of the number of polyps in the entire intestine of DBZ or vehicle-treated *Lgr5^CreERT2 Apc^fl/fl* mice (NIC-treated and untreated). (**F**) Representative images (Red: β-catenin, Blue: DAPI) and the quantification of the number of β-catenin positive adenomatous lesions in the entire intestine of DBZ or vehicle-treated *Lgr5^CreERT2 Apc^fl/fl* mice (NIC-treated and untreated). Original magnifications: ×200 (C, and F). Scale bar: 50 μm (C, and F). Values represent the mean ± SEM. Significant differences are denoted by p values (Student's t-test).

The online version of this article includes the following source data for figure 7:

**Source data 1.** The quantification of polyps and β-catenin positive lesions.

Industry CO., LTD, Tokyo, Japan), and Rapamycin (LKT Laboratories, Inc, St. Paul, MN), was added to the culture medium as needed in different experiments. The absolute values of the organoids are plotted on the y-axis in *Figure 2* and S2. The ratio to the control is presented on the y-axis in *Figures 3–5*. The number of colonies with lumens was quantitated on days 3 (*Figure 2—figure supplement 1A*) and 5 (*Figures 2–5*) of the culture.

## Colonic crypt isolation and culture

Colonic crypts were isolated from the large intestine as described previously with a few modifications[23]. For this purpose, a 5–7 cm part of the proximal large intestine was isolated, opened longitudinally, and washed with cold PBS. After washing with cold PBS, the intestine was cut into small (5 mm long) pieces with scissors, placed in cold 5 mM EDTA-PBS, and gently rocked for 15 min at 4 °C. After removal of EDTA-PBS, pieces of the intestine were incubated in DMEM/F12 containing 500 U/ml Collagenase Type IV (Worthington Biochemical Corporation, Lakewood, NJ) for 30 min at 37 °C using a water bath. Subsequently, pieces of the intestine were pipetted up and down in cold PBS until most of the crypts were released. The crypt fraction was obtained by passing the suspension through a 70 μm cell strainer followed by centrifuging at 250 × g for 5 min. Isolated crypts were collected in a crypt culture medium, counted, and embedded in Matrigel. A total of 1000 crypts were plated per well of a 48-well plate and cultured using a colonic crypt culture medium (DMEM/F12 supplemented by 1xN2, 1xB27, 1 mM N-Acetyl-L-cysteine, 50 ng/ml EGF, 100 ng/ml Noggin, 500 ng/ml R-spondin, 2 mM Valproic acid (FUJIFILM Wako Pure Chemical Corporation), and 10 μM CHIR99021). The medium was replaced in 2 days.

Nicotine hemisulfate salt was added to the crypt culture medium as needed for experiments. The number of alive spherical organoids formed from the crypts was measured under microscope 5 days after plating them.

## Investigation of intestinal polyps

The entire intestine of *Lgr5^CreERT2 Apc^fl/fl:tdTomato* mice, in which adenomatous polyps were labeled with tdTomato using tamoxifen injection, was promptly excised and cut with the mucosal side up, washed with ice-cold PBS, pinned open on a dissection tray, and fixed using 10% neutral buffered formalin (FUJIFILM Wako Pure Chemical Corporation). The fixed intestine was then photographed, polyps were counted, their diameters were measured using a caliper, and the surface of the polyps was estimated. For the histopathological assay, the entire intestine was fixed on a dry board and gently rolled to form a Swiss roll, which was further used in the immunohistological test.

## Immunohistochemistry

Pieces of the proximal jejunum (1–4 cm from the pylorus), proximal colon (1–2 cm from the cecum), and the entire rolled intestine were fixed overnight using 10% neutral-buffered formalin, embedded in paraffin, and sections were prepared. The sections were deparaffinized and subjected to heat-induced antigen retrieval using 10 mM sodium citrate buffer (pH 6.0) in a microwave. Slides were then incubated overnight with the following primary antibodies at 4 °C: rabbit anti-Ki67 (1/200; 12202; Cell Signaling Technology, Danvers, MA), rabbit anti-Olfm4 (1/400; 39141, Cell Signaling Technology), mouse anti-GFP (1/50; sc-9996, Santa Cruz Biotechnology, Dallas, TX), rabbit anti-Lysozyme (1/50; PA5-16668; Thermo Fisher Schientific), mouse anti-chromogranin A (ChgA; 1/50; sc-393941, Santa Cruz Biotechnology), and rabbit anti-β-catenin (1/200; 9562; Cell Signaling). For samples incubated with mouse primary antibodies, Alexa Fluor-conjugated secondary antibody (1/200; ab150113;

Abcam, Cambridge, UK) were used, whereas, a TSA Plus Cyanine 3 System (Akoya Biosciences, Marlborough, MA,) was used for rabbit primary antibody treated sets following instructions provided by the manufacturer. Finally, the slides were mounted using Mounting medium With DAPI-Aqueous, Fluoroshield (Abcam, Cambridge, MA). Images were recorded using an all-in-one fluorescence Microscope APEXVIEW APX100 (Olympus,Tokyo, Japan).

### Quantification

The lengths of the crypt (from the bottom of the crypt to the crypt-villus junction) and villus (from the crypt-villus junction to the tip of the villus) were measured using ImageJ software. The quantification was repeated for >30 crypt/villus units per mouse. Immuno-stained cells were quantified using randomly selected 50 intact, well-oriented crypts per mouse.

### Immunoblotting

The following antibodies, obtained from different sources, were used for immunoblotting: mouse anti-β-actin (sc-47778; Santa Cruz), mouse anti-YAP (Santa Cruz sc-101199), mouse anti-TAZ (Santa Cruz sc-293183), mouse anti-α7-AchR (Santa Cruz sc-58607), mouse anti-Notch1 (Santa Cruz sc-376403), mouse anti-Jagged1 (Santa Cruz sc-390177), mouse anti-Jagged2 (Santa Cruz sc-515725), mouse anti-Hes5 (Santa Cruz sc-293445), mouse anti-p38 (Santa Cruz sc-81621), mouse anti-phospho-p38 (Santa Cruz sc-166182), rabbit anti-S6 (Cell Signaling 2217), and rabbit anti-phospho-S6 Ser235/236 (Cell Signaling 4858). Crypts or sorted cells were lysed using RIPA buffer supplemented with protease and phosphatase inhibitors (Santa Cruz Biotechnology). Subsequently, the protein extracts were denatured by adding SDS loading buffer, boiled, and resolved using SDS-PAGE; immunoblotting was performed using primary antibodies listed above. Mouse IgG (NA931; Cytiva, Tokyo, Japan) and rabbit IgG (NA934; Cytiva) antibodies were used as HRP-conjugated secondary antibodies. The band signals were enhanced using Can Get Signal Immunoreaction Enhancer Solution (TOYOBO CO., LTD., OSAKA, Japan) following the instructions provided by the manufacturer. The band density of all blots was quantified by Image J software.

### RNA analysis by real-time qPCR

RNA was extracted from crypts or sorted cells using the RNeasy Plus Mini Kit (QIAGEN, Hilden, Germany). Reverse transcription was performed using ReverTra Ace qPCR RT Master Mix (TOYOBO CO., LTD.). qRT-PCR was performed on a QuantStudio 5 Real-Time PCR System (Applied Biosystems, Waltham, MA) using KAPA SYBR FAST qPCR Master Mix (Kapa Biosystems, Inc, Wilmington, MA). Sequences of the primers used for the Real-Time qPCR are listed below. 18 S rRNA was considered an endogenous reference gene.

### The list of primers used for real-time qPCR

|      | Forward primer | Reverse Primer |
|------|----------------|----------------|
| α1   | TATAACAACGCAGACGGCGA | CACAGAGACCGTCATAGGTCC |
| α2   | GTGCCCAACACTTCCGATG | TGTAGTCATTCCATTCCTGCTTT |
| α3   | CCAGTTTGAGGTGTCTATGTC | TCGGCGTTGTTGTAAAGC |
| α4   | CTCAGATGTGGTCCTTGTC | GAGTTCAGATGGGATGCG |
| α5   | CATCGTTTTGTTTGATAATGC | TGCGTCCAAGTGACAGTG |
| α6   | TGTCTCCGATCCCGTCAC | TTGTTATACAGAACGATGTCAGG |
| α7   | GGTCATTTGCCCACTCTG | GACAGCCTATCGGGTGAG |
| α9   | ACAAGGCCACCAACTCCA | ACCAACCCACTCCTCCTCTT |
| α10  | TCTGACCTCACAACCCACAA | TCCTGTCTCAGCCTCCATGT |
| β1   | AAGCCGAAGGCCAACTGATTA | TCCTGCCTCTCCTCTCCTTC |
| β2   | CCGGCAAGAAGCCGGGACCT | CTCGCTGACACAAGGGCTGCG |
| β3   | AAGAAGCAGACTCCTACC | AACAACCTGACTGATGAAG |

*Continued on next page*

*Continued*

|  | Forward primer | Reverse Primer |
|---|---|---|
| β4 | CTACAGGAAGCATTAGAGG | CAGAATACACACAATCACG |
| Hes1 | ACACCGGACAAACCAAAGAC | AATGCCGGGAGCTATCTTTC |
| Hes5 | GCAGCATAGAGCAGCTGAAG | AGGCTTTGCTGTGTTTCAGG |
| HeyL | GTCTTGCAGATGACCGTGGA | CTCGGGCATCAAAGAACCCT |
| Hey1 | CACCTGAAAATGCTGCACAC | ATGCTCAGATAACGGGCAAC |
| Hes1 | ACACCGGACAAACCAAAGAC | AATGCCGGGAGCTATCTTTC |
| Jagged1 | CCTCGGGTCAGTTTGAGCTG | CCTTGAGGCACACTTTGAAGTA |
| Jagged2 | ACGAGGAGGATGAAGAGCTGA | GGGGTCTTTGGTGAACTTGTG |
| YAP | CGCTCTTCAATGCCGTCATG | AGTCATGGCTTGCTCCCATC |
| TAZ | TCTGTCATGAACCCCAAGCC | GGTGGTTCTGTGGACTCAGG |
| 18 S | GTAACCCGTTGAACCCCATT | CCATCCAATCGGTAGTAGCG |

## Single-cell gene expression analysis

We used single-cell RNA-seq data from the paper 'Cells of the human intestinal tract mapped across space and time' published in Nature (*Elmentaite et al., 2021*) for our analysis. This mapped count data was used as the input for data processing with the Seurat R package (version 5.0.3; *Hao et al., 2024*). We used the 'FindVariableFeatures' function to identify highly variable features for downstream analysis. This data was used for dimensionality reduction and cluster detection. We performed linear regression using the 'ScaleData' function and a linear dimensionality reduction using the 'RunPCA' function. Twenty principal components were used for downstream graph-based, supervised clustering into distinct populations using the 'FindClusters' function and uniform manifold approximation, and projection (UMAP) dimensionality reduction was performed to project the cell population onto two dimensions using the 'RunUMAP' function. We used this UMAP data to compare CHRNA7 with other receptor subunits and to verify differences in their distribution.

## Statistical analysis

All experiment as indicated were performed by at least three independent experiments or technical replicates. Quantitative results are presented as mean ± standard error of the mean. Two groups were compared using unpaired two-tailed Student's t-test, assuming data normality. p was set at $p < 0.05$ significant.

## Acknowledgements

Flow cytometry was performed in the IMSUT FACS Core laboratory. We acknowledge the IMSUT FACS Core laboratory for assistance with flow cytometry analysis. We thank Dr Yoku Hayakawa (The University of Tokyo) for providing Apc[flox/flox] mouse. Finally, we would like to thank Editage (https://www.editage.com/) for the English language editing. MI was supported by the Smoking Research Foundation.

## Additional information

### Funding

| Funder | Grant reference number | Author |
|---|---|---|
| Smoking Research Foundation |  | Masaki Igarashi |

The funders had no role in study design, data collection and interpretation, or the decision to submit the work for publication.

## Author contributions
Ryosuke Isotani, Conceptualization, Data curation, Formal analysis, Investigation, Visualization, Methodology, Writing – original draft, Project administration; Masaki Igarashi, Conceptualization, Resources, Data curation, Formal analysis, Supervision, Funding acquisition, Validation, Investigation, Visualization, Methodology, Writing – original draft, Project administration, Writing – review and editing; Masaomi Miura, Data curation, Formal analysis, Investigation, Project administration; Kyoko Naruse, Satoshi Kuranami, Resources; Manami Katoh, Software, Formal analysis, Investigation, Visualization, Methodology, Writing – original draft, Project administration, Writing – review and editing; Seitaro Nomura, Formal analysis, Supervision, Investigation, Methodology, Project administration; Toshimasa Yamauchi, Conceptualization, Supervision, Validation

## Author ORCIDs
Masaki Igarashi ⓘ https://orcid.org/0000-0002-3331-3877

## Ethics
This study was performed following the guidelines of the Animal Care Committee of the University of Tokyo. All of the animals were handled according to the protocol approved by the Committee on the Ethics of Animal Experiments of the University of Tokyo (Permit Number: A2023M043-04).

Reviewer #1 (Public review): https://doi.org/10.7554/eLife.95267.4.sa1
Reviewer #2 (Public review): https://doi.org/10.7554/eLife.95267.4.sa2
Author response https://doi.org/10.7554/eLife.95267.4.sa3

---

# Additional files

### Supplementary files
MDAR checklist

### Data availability
All data generated or analyzed during this study are included in the manuscript and supporting files; source data files have been provided for Figures 1-7, Figure 1-figure supplement 1, Figure 2-figure supplement 1, Figure 4-figure supplement 1 and Figure 6-figure supplement 1.

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
