## [Editor Report · eLife Assessment]

This study presents a **valuable** finding on a potential signaling pathway responsible for the direct effects of nicotine on intestinal stem cell growth and tumorigenesis. The evidence supporting the claims of the authors is **solid**. This research will be of interest to medical biologists specializing in intestinal tumors.

---

## [Referee Report · Reviewer #1 (Public review)]

In their manuscript, authors Isotani et al used in vivo and ex vivo models to show that nicotine could promote stemness and tumorigenicity in murine model. The authors further provided data supporting that the effects of nicotine on stem cell proliferation and tumor initiation were mediated by the Hippo-YAP/TAZ and Notch signal pathway.

The major strength of this study is the using a set of tools, including Lgr5 reporter mice (Lgr5-EGFP-IRES-CreERT2 mice), stem cell-specific Apc knockout mice (Lgr5CreER Apcfl/fl mice), organoids derived from these mice and chemical compounds (agonists and antagonists) to demonstrate nicotine affects stem cells rather than Paneth cells, leading to increased intestinal stemness and tumorigenicity. Whereas, all models are restricted to mice, lacking analysis of human samples or human intestinal organoids to prove the human relevance of these findings.

Overall, the presented results support their conclusions. A previous study reported that nicotine acts through the α2β4 nAChR to enhance Wnt production by Paneth cells, which subsequently affects ISCs. In contrast, this manuscript demonstrated that nicotine directly promotes ISCs through α7-nAChR, independent of Paneth cells. Therefore, this manuscript offers novel insights into the mechanism of nicotine's effects on the mouse intestine.

---

## [Referee Report · Reviewer #2 (Public review)]

Summary:

The manuscript by Isotani et al characterizes the hyperproliferation of intestinal stem cells (ISCs) induced by nicotine treatment in vivo. Employing a range of small molecule inhibitors, the authors systematically investigated potential receptors and downstream pathways associated with nicotine-induced phenotypes through in vitro organoid experiments. Notably, the study specifically highlights a signaling cascade involving α7-nAChR/PKC/YAP/TAZ/Notch as a key driver of nicotine-induced stem cell hyperproliferation. Utilizing a Lgr5CreER Apcfl/fl mouse model, the authors extend their findings to propose a potential role of nicotine in stem cell tumorgenesis. The study posits that Notch signaling is essential during this process.

Strengths and Weaknesses:

One noteworthy research highlight in this study is the indication, as shown in Figure 2 and S2, that the trophic effect of nicotine on ISC expansion is independent of Paneth cells. In the Discussion section, the authors propose that this independence may be attributed to distinct expression patterns of nAChRs in different cell types. To further substantiate these findings, the authors provided qPCR analysis of nAchRs in ISCs and Paneth cells from isolated whole small intestine, indicating that α7-nAChR uniquely responds to nicotine treatment among various nAChRs. The authors further strengthen the clinical relevance of the study by exploring human scRNA-seq dataset, in which α7-nAChR is indeed also expressed in human ISCs and Paneth cells.

As shown in the same result section, the effect of nicotine on ISC organoid formation appears to be independent of CHIR99021, a Wnt activator. In the Lgr5CreER Apcfl/fl mouse model, it is known that APC loss results in a constitutive stabilization of β-catenin, thus the hyperproliferation of ISCs by nicotine treatment in this mouse model is likely beyond Wnt activation. The authors have included such discussion.

In Figure 4, the authors investigate ISC organoid formation with a pan-PKC inhibitor, revealing that PKC inhibition blocks nicotine-induced ISC expansion. It's noteworthy that PKC inhibitors have historically been used successfully to isolate and maintain stem cells by promoting self-renewal. Therefore, it is surprising to observe no or reversal effect on ISCs in this context. The authors have now included an additional PKC inhibitor Sotrastaurin to confirm the role of PKC in nicotine-induced ISC expansion.

Overall, the manuscript has provided sufficient experimental evidence to address my concerns and also significantly enhanced its quality.

---

## [Author Response]

The following is the authors’ response to the previous reviews.

**Public Reviews:**

**Reviewer #1 (Public Review):**
Strengths and weaknesses:Although the revised manuscript has significantly improved in the quality of pictures, there seems to be still a discrepancy in Figure 2A: quantification result suggested that NIC (1um) treatment increased the number of colonies from 300 to around 450 (1.5 folds), whereas representative picture shown that the difference was 3 to 12 living organoids (4 folds).

As reviewer points out, the selected picture was not representative image of “control” group in Figure2A. We replaced it by the new representative image in this revised version.

**Recommendations for the authors**:
**Reviewer #2 (Recommendations for the authors):**
A minor point to be corrected:Please consider removing "In consistent with this notion", which is repetitive with "Similarly"." NIC is supposed to activate Wnt signaling via Hippo-YAP/TAZ and Notch signaling. In consistent with this notion. Similarly, the expression of target proteins (Sox9, TCF4 and, C-myc)..."

We corrected it according to the reviewer’s suggestion.